computer modelling and simulation/statistical physics/artificial intelligence

evolution of cooperation, exploration dynamics, social networks

**Author for correspondence:**
Flávio L. Pinheiro
e-mail: fpinheiro@novaims.unl.pt

# Stable leaders pave the way for cooperation under time-dependent exploration rates

Flávio L. Pinheiro[1,5], Jorge M. Pacheco[2,3,5] and Francisco C. Santos[4,5]

[1]NOVA Information Management School (NOVA IMS), Universidade Nova de Lisboa, Campus de Campolide, 1070-312 Lisboa, Portugal
[2]Centro de Biologia Molecular e Ambiental, and [3]Departamento de Matemática e Aplicações, Universidade do Minho, 4710-057 Braga, Portugal
[4]INESC-ID and Instituto Superior Técnico, Universidade de Lisboa, 2744-016 Porto Salvo, Portugal
[5]ATP-group, 2744-016 Porto Salvo, Portugal

FLP, 0000-0002-0561-9641; JMP, 0000-0002-2579-8499; FCS, 0000-0002-9103-2862

The exploration of different behaviours is part of the adaptation repertoire of individuals to new environments. Here, we explore how the evolution of cooperative behaviour is affected by the interplay between exploration dynamics and social learning, in particular when individuals engage on prisoner's dilemma along the edges of a social network. We show that when the population undergoes a transition from strong to weak exploration rates a decline in the overall levels of cooperation is observed. However, if the rate of decay is lower in highly connected individuals (Leaders) than for the less connected individuals (Followers) then the population is able to achieve higher levels of cooperation. Finally, we show that minor differences in selection intensities (the degree of determinism in social learning) and individual exploration rates, can translate into major differences in the observed collective dynamics.

## 1. Introduction

Humans often replicate the behaviours of others that exhibit higher social fitness, and do so by means of social learning and peer influence [1,2]. However, individuals can also unilaterally adopt new behaviours, e.g. through learning errors or innovations [3]. This way, populations are capable to explore behaviours otherwise inaccessible to them [4,5]. Hence, when studying population dynamics, it is usually assumed that the evolution of individual behaviours in socio-economic systems proceeds, in many respects, in a way that mimics the fundamental forces driving its biological counterpart [6–9], that is, through a

combination of both selection and mutation (exploration). Here, we study the impact of exploration dynamics in the evolution of cooperative behaviour, framed within the context of evolutionary game theory. In particular, we explore the implications, on the population-wide dynamics, of introducing a finite exploration rate, as well as how the collective dynamics is affected when a transition from strong to weak exploration rates takes place.

Past literature in evolutionary game theory typically assumes scenarios in which both selection pressure and exploration rates are constant and homogeneous across the population [10–17] some exceptions include Szolnoki *et al.* [18,19] and Alam *et al.* [20]. Arguably, a more realistic scenario would consider time varying exploration rates, which can be triggered by external shocks or internal dynamics. For instance, it is known that human learners often adopt a high exploration rate at earlier stages when facing a new environment, progressively reducing such rate in time [21], a feature which allows for a better exploration of the available strategy space [22–24]. It has been also shown that some humans are very selective about when to use information from third parties, which leads to a heterogeneous distribution of learning strategies [25]. Moreover, several findings in biology [26–28] seemingly support the idea that populations can undergo variations in mutation rates as a response to environmental pressures. Such findings naturally raise questions on the impact of variable exploration rates in the evolution of cooperative behaviour, in particular on structured populations [10,12,16,29,30] and when individuals engage in prisoner's dilemma (PD) [6,31].

The emergent population-wide evolutionary dynamics in structured populations can be strikingly different from what one would expect from the dilemma played in well-mixed populations [11,32]. For instance, when individuals are connected through a heterogeneous social network, the dynamics associated with a PD will be transformed into a coordination dilemma game [11,32], in sharp contrast with the defection dominance dynamics related to the pay-off structure of the game and how the game is locally perceived. In that sense, and because of the above properties of interaction structures in population dynamics, these structures have been suggested as one of the fundamental evolutionary mechanisms supporting the emergence of cooperative behaviour in populations of rationally bounded individuals [33]. But what happens when we take into account exploration dynamics? Recently, it was shown that a similar, but more complex, break in symmetry between individual-level and the emergent population-wide dynamics happens [34,35]. But what does it imply to populations that might undergo time-dependent variations in exploration rates? In that context, past research provides some insights by showing that the emergence of cooperation can be favour when individual's social learning ability lowers with ageing [36].

Several empirical works have confirmed the positive impact of population structure in the evolution of cooperation [37–39]. However, in other instances [4,40,41] it was reported that some individuals would spontaneously change their strategy to defection, leading to the collapse of cooperation. An explanation proposed for such phenomena [38,40] involves the effect of 'mutation/exploration dynamics' (the individuals' willingness to explore the strategic space) as it affects the creation of the strategy assortments (i.e. stable clusters of cooperators) necessary for cooperation to thrive on networks. In other words, existing and unaccounted exploration rates—which can be due to the experimental set-up, stochastic effects, inconsistent individual's strategic preferences and the population of subjects, among others—may lead to the reported mismatch between theory and empirical evidence. Naturally, this is an effect that affects all experiments, thus its relevance. Interestingly, recent results have shown that such 'noisy' players can also be vital for the promotion of cooperative behaviour. By strategically placing artificial agents with noisy behaviours (i.e. high exploration rates) in different locations of a social network, it was shown that such agents would facilitate the solving of complex tasks by a population of human participants [42]. Moreover, the selection pressure—i.e. how important the game pay-off is for the individual fitness—may also have a strong influence in decision-making in networked populations [15,32,40]. Thus, our work also aims at providing insights on the interplay between exploration rates, population structure and other sources of stochasticity, such as selection pressure, that could, e.g. lead to a better understanding of experimental results.

## 2. Results

We start by showing that the exploration rate has a profound impact on the effective dilemma individuals face. Secondly, we show that when facing a variation in the exploration rates, from strong to weak, populations can reach undesirable outcomes in the level of cooperation—for instance, when comparing with situations where exploration dynamics is very low or absent. Such dynamical

outcomes are not possible to reproduce in well-mixed and homogeneously structured populations (see Methods). We further investigate how by decoupling the rate at which exploration levels vary between Leaders (highly connected individuals) and Followers (lower connected individuals) impacts the overall levels of cooperation. We show that allowing Leaders to converge faster to low exploration rates can benefit the evolution of cooperative behaviour. For simplicity, and since the network structures are static, individuals are assigned to either a role of Leader or Follower at the start of the evolution, depending on their number of contacts (degree), and remain in that role for the entirety of the simulation.

Let us consider that interactions among individuals occur in terms of the two-person PD, where players can either cooperate or defect during an interaction. Mutual cooperation leads to a reward, $R$, whereas mutual defection leads to a punishment $P$ ($<R$). When one player cooperates and the other defects, the associated game pay-offs are, respectively, $S$ (sucker's pay-off) and $T$ (temptation). The PD game is obtained when $T > R > P > S$, such that mutual cooperation ($R$) is preferred over unilateral cooperation ($S$) and mutual defection ($P$). Nonetheless, defection tends to be chosen due to the combination of two social tensions [43]: the temptation to cheat against a cooperator ($T$ is the best possible outcome), and due to the fear of being cheated ($S$ is the worst possible return). The ensuing pay-off structure and further details can be found in the Methods.

Although popular, the evolutionary dynamics of a finite population in the well-mixed approximation usually constitutes a strong assumption that is seldom realized. In some simple cases, one may have populations spatially and quasi-regularly distributed [12]. More often, population structures exhibit sizable irregularities, which translate into a structure well represented by heterogeneous graphs, often exhibiting fat tails whenever populations are sufficiently large [30] and heterogeneity in what concerns the distribution of the number of interactions [44,45]. Hence, in the remainder of the manuscript we probe the impact of variable exploration levels on degree heterogeneous structured populations [11] with a scale-free degree distribution, as representative example of such heterogeneous graphs. We have generated these networks using the Barabási–Albert algorithm of growth and preferential attachment [30] (see Methods), in this way capturing some of the features of real-world networks detailed above. Given this setting, we shall now address how the emergent macroscopic dynamics characterized in structured populations is altered for different exploration rates and selection pressures.

Behavioural dynamics is modelled via social learning, through the so-called pairwise comparison rule [46]. Here, an individual $i$ with fitness $f_i$ imitates the strategy of a randomly chosen neighbour $j$ (with fitness $f_j$) with a probability that increases with the fitness difference ($\Delta_f = f_j - f_i$). Such probability can be given by the Fermi distribution $p = [1 + e^{-\beta \Delta_f}]^{-1}$, where $\beta$ provides a convenient measure of the strength of natural selection and the errors associated with the social learning process. The overall dynamics in well-mixed populations (i.e. a complete graph) can be described analytically by means of the gradient of selection $G(k) = T^+(k) - T^-(k)$, representing the difference between the probabilities of increasing ($T^+(k)$) and decreasing ($T^-(k)$) the number of cooperators by one when there are $k$ cooperators in the population. The internal roots of $G(k)$ provide finite-population analogues of internal fixed points in infinite populations [31,47–51]. The addition of exploration prompts the emergence of an additional internal fixed point $x_L$ that emerges and move towards $x = 0.5$ with increasing $\mu$ (see orange lines in figure 1). In other words, exploration dynamics trivially transforms a purely defection-dominance dilemma into a coexistence game. In such cases, the final balance of cooperators and defectors is dictated by the location of the attractor $x_L$ (a finite population analogue of a stable fixed point in infinite populations).

This description is, however, too simple to account for the complex dynamics emerging from structured populations [3,11,29,35,52,53]. Alternatively, we resort to the average gradient of selection, $\Gamma_\mu(k, t)$, which is a numerically computed counterpart of $G(k)$ [11,52,53]. Unlike $G(k)$, $\Gamma_\mu(k, t)$ (where $\mu$ is the exploration rate and $t$ is the time) is a time-dependent quantity that allows to track the self-organization of cooperators and the effective game being played at different times. Moreover, the average gradient of selection (AGoS) can be computed for arbitrary intensity of selection, population structure and game parametrization. Past works have used the AGoS to link individual to the emergent population-level dynamics, showing that different types of social networks lead to different population-wide dynamics [53].

Figure 1 shows the location of the internal roots of the average gradient of selection strongly depends on the exploration rates. We picture the value of $\Gamma_\mu(k, g)$ for a broad range of (fixed) exploration rates ($10^{-4} \leq \mu \leq 1.0$) and for the case with $T = 1.25$ and $S = -0.25$ (PD domain with $R = 1$ and $P = 0$). Each panel of figure 1 shows the results obtained for different selection pressures $\beta$ (see Methods): $\beta = 0.01$ ($a$), $\beta = 0.1$ ($b$) and $\beta = 10.0$ ($c$).

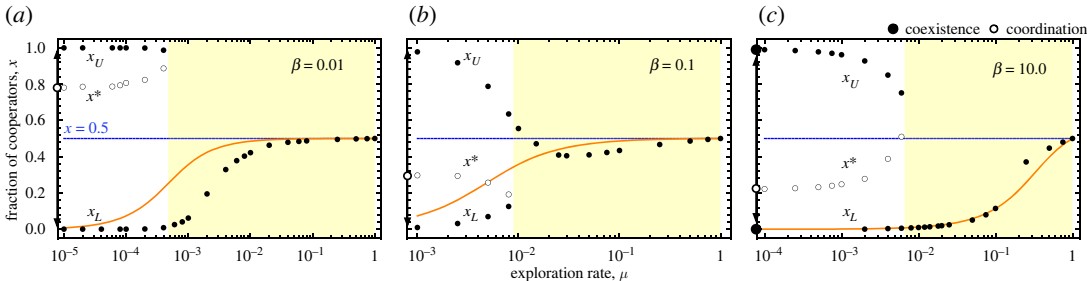

**Figure 1.** Internal roots of the average gradient of selection $\Gamma_\mu(k, g)$ at $g = 100$ generations (see Methods) on heterogeneous structured populations as a function of the exploration probability $\mu$. Each panel shows results for different selection pressures: (a) $\beta = 0.01$, (b) $\beta = 0.1$ and (c) $\beta = 10.0$. The yellow background denotes the region $\mu > \mu_C$, the value at which two internal roots (the finite-population analogues of fixed points in infinite populations) of $\Gamma_\mu(k, g)$ coalesce. When $\mu < \mu_C$ the evolutionary outcome is dictated by the coordination point ($x^*$) (i.e. a repeller, a finite population analogue of an unstable fixed point), meaning that depending on the initial fraction of cooperators the population could be driven towards any of the two possible basins of attraction. When $\mu > \mu_C$, evolution drives the population towards a coexistence, identified by the location of the attractor (a finite population analogue of a stable fixed point), that is dictated by the strength of the exploration dynamics. Results for $\mu = 0.0$ (i.e. in absence of exploration dynamics), are represented in the left border of each panel. In such case the dynamics is characterized by a single repeller ($x^*$). Orange lines depict the location of the internal coexistence point that characterizes the evolution of well-mixed populations under exploration dynamics (see Methods for a detailed discussion of the results for well-mixed populations). Other parameters are $T = 1.25$, $S = -0.25$, $Z = 10^3$ and network average connectivity $\langle k \rangle = 4$.

For $\mu = 0.0$ and $\beta = 0.01$ (figure 1a) or $\beta = 0.1$ (figure 1b) the population-wide dynamics is dominated by a single internal repeller at $x^*$ (a finite population analogue of an unstable fixed point in infinite populations), which implies the population will evolve under a coordination dynamics and be driven to either $x = 1.0$ or $x = 0.0$ depending on its initial condition. For stronger selection ($\beta = 10$, figure 1c), additional roots appear in the close vicinity of $x = 1.0$ and $x = 0.0$ which, while playing no relevant role in the present analysis, result from specific assortments of strategies which lead, in the absence of exploration, to long fixation times [32]. The coordination nature of the population-wide dynamics, observed for $\mu = 0.0$, implies that an increasing $\mu$ leads to the emergence of two internal probability attractors near $x = 1.0$ ($x_U$) and $x = 0.0$ ($x_L$), both approaching $x = 0.5$ with increasing $\mu$.

From figure 1, one may easily infer that, as in many other complex adaptive systems [7,54–57], evolution in structured populations portrays critical thresholds (sometimes called tipping points) at which the population-wide dynamics abruptly shifts from one regime to another. In this case, two distinct dynamical pictures result with increasing exploration probabilities ($\mu$) which also depend on the selection pressure. For $\beta = 0.1$ (figure 1b), $x^*$ coalesces with $x_L$, whereas for $\beta = 0.01$ and $10.0$ (figure 1a and 1c) $x^*$ coalesces with $x_U$. In the present scenario, and since for $\mu = 0.0$ the population-wide dynamics is characterized by single repeller, with increasing $\mu$ we will always observe the emergence of two additional internal roots: $x_U$ and $x_L$. One of these additional roots will coalesce with the repeller, the level of $\mu$ at which the coalescence happens depends on the specific conditions of the simulation. Interestingly, whenever selection is strong (figure 1c), above a critical exploration rate, evolution proceeds as if the network is absent, as we abruptly recover the well-mixed dynamical profile with a single coexistence internal point (see orange lines). For lower selection pressures (figure 1a,b), such critical exploration rate defines the point above which population structure and degree heterogeneity become detrimental to cooperation when compared with populations without any interaction structure.

Figure 1 also suggests that, unlike the scenario associated with well-mixed populations, the evolutionary outcome under variable exploration rates on structured populations will also depend on the overall selection pressure: A parametric increase of the exploration rate will drive populations towards a coexistence of strategies, regardless of both (i) initial configuration of the population and (ii) selection pressure. However, a time-dependent decrease in the exploration rate implies the demise of cooperation for a wide range of selection pressures ($\beta < 0.1$ and $\beta > 0.5$, see figure 2a). Between these domains (more precisely, $0.1 < \beta < 0.5$) the overall dynamics is dominated by an interior attractor, already when $\mu \ll 1$. These results highlight the existence of an optimal selection pressure at which cooperation is maximized [32].

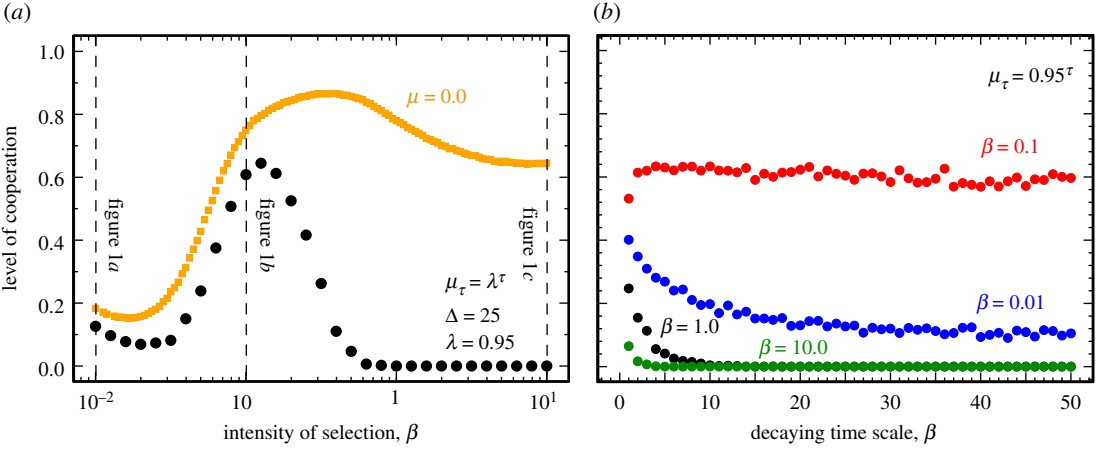

**Figure 2.** Level of cooperation under variable, time-dependent, explorations for a wide range of selection pressures (*a*) and decaying time scales (*b*) on scale-free networks. Panel (*a*) shows the average final fraction of cooperation under variable exploration (black discs) when compared with the level of cooperation attained in a scenario without exploration ($\mu = 0$, orange squares). Vertical dashed lines indicate the intensity of selection ($\beta$) values used in figure 1. Panel (*b*) shows the average final fraction of cooperation for different decaying time scales of the exploration rate for different selection pressures. Other parameters are $S = -0.25$, $T = 1.25$, $Z = 10^3$, $\lambda = 0.95$ and $\langle k \rangle = 4$.

Let us now induce the population to undergo a transition from strong ($\mu = 1.0$) to weak ($\mu = 10^{-5}$) exploration regimes. Let us start from a random composition of strategies at $\mu_0 = 1.0$. After each period $\tau$ (with a duration of $\Delta$ generations (see Methods) the population undergoes a decrease in $\mu$ by a fraction $\lambda < 1.0$: $\mu_{\tau+1} = \lambda \mu_\tau$, this process being repeated while $\mu_\tau > 10^{-5}$. Figure 2*a* shows the level of cooperation (see Methods) obtained this way with black dots. For comparison, orange squares depict the corresponding values in the limit $\mu = 0$. We span three orders of magnitude of values for the selection pressure $\beta$ at constant $\lambda = 0.95$. Figure 2*b*, in turn, shows the dependence of the level of cooperation on the decaying time-scale value $\Delta$ adopted, for different selection pressures.

Clearly, variable exploration rates (as described above) will always lead to lower levels of cooperation (compared with the $\mu = 0$ case) which, for $\beta > 0.5$, result in the complete demise of cooperation. For lower selection pressures, the stochastic effects compensate this harsher dynamical picture (also evidenced in figure 1*a*). Moreover, as shown in figure 2*b*, the decrease in cooperation becomes more pronounced for larger $\Delta$, that is, in regimes where the exploration rates decay slowly. Finally, in line with the static results of figure 1, there is a window of hope for cooperators within a range of selection pressures (around $\beta = 0.1$) which optimizes the resilience of populations to changes of exploration rates. In what concerns the impact of different decay time scales (figure 2*b*), when $\beta = 0.01$ the observed level of cooperation remains stable already for $\Delta > 5$, while for increasing selection pressures they tend to stabilize at higher values of $\Delta$.

The previous results present somewhat grim prospects for cooperation. Regardless of whether exploration is static or time dependent, the population will always achieve a lower level of cooperation when compared with situations where no exploration takes place. However, it is important to keep in mind that, in all cases, we considered 'uniform populations' in which all individuals explored the fitness landscape at the same rate. This may not always be the case. Indeed, individuals' willingness to explore can be conditional on their level of social influence, or on how many social ties they hold. In the following, we investigate the consequences of this possibility by letting individuals with different number of neighbours adopt distinct exploration rates. To this end, we split the population in two classes: Leaders who have a degree greater than 2/3 of the maximum degree of the population (highly connected individuals), and Followers. Figure 3*a* shows how a decoupling of the exploration rates between such two classes—Leaders and Followers—impacts the level of cooperation. In order to highlight how these results deviate from a baseline condition we show the relative increase in the level of cooperation compared with the baseline scenario where no exploration takes place ($\mu = 0$). Results worse than the baseline are shown in black; results better than the baseline are indicated with brighter colours, the more bright the larger the positive deviation. We fix the selection pressure to an intermediate value ($\beta \approx 0.1$). Clearly, Followers with exploration rates higher than Leaders may favour the increase of

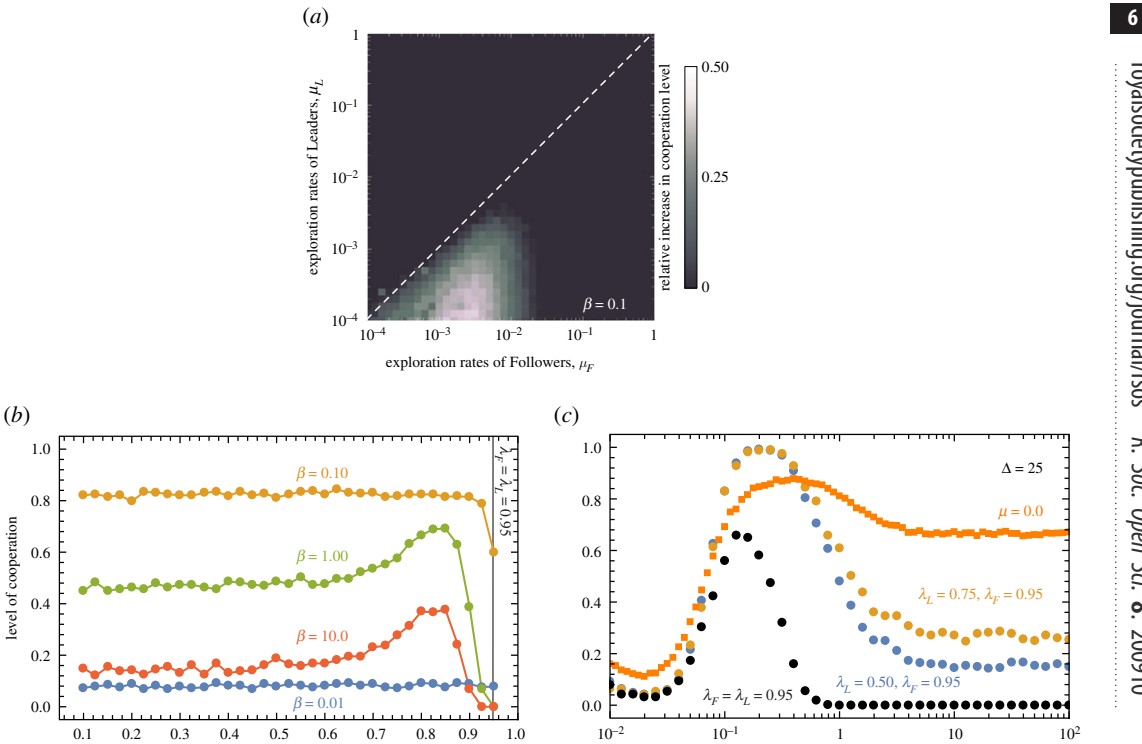

**Figure 3.** Level of cooperation under asymmetric exploration rates. In panel (*a*) the *Y*-axis represents the exploration rates of Leaders, here defined as individuals with a degree greater or equal than 2/3 of the maximum degree in the population, while the *X*-axis identifies the exploration rates of Followers, individuals with a degree lower than 2/3 the maximum degree in the population. Results depict the gains in cooperation compared with a baseline scenario where no exploration takes place. Black regions indicate situations in which the level of cooperation achieved is equal or worse than the baseline, while brighter tones indicate conditions in which the level of cooperation was greater than the baseline, the brighter the greater. Panel (*b*) shows how the levels of cooperation depend on the balance between the exploration rates of the Leaders ($0.1 \leq \lambda_L \leq 0.95$) and the Followers ($\lambda_F$). In panels (*a*) and (*b*), exploration rates are constant, not time-dependent, and set at the start of each simulation. Panel (*c*) shows how the levels of cooperation depend on the interplay between $\lambda_L$, $\lambda_F$ and selection pressure ($\beta$) when exploration rates are time-dependent ($\Delta = 25$). Other parameters, $S = -0.25$, $T = 1.25$, $Z = 10^3$ and $\langle k \rangle = 4$.

cooperation levels by as much as 50%. This however, only occurs for intermediate selection pressure, as for both strong and weak selection, the outcomes are always worse when compared with their baseline counterpart.

In the light of the previous results, it becomes relevant to investigate to which extent the levels of cooperation are sensitive to a decay of the exploration rates that also depends on the connectivity of individuals. To this end, let us assume, as before, that the population starts from uniform and strong exploration rates ($\mu = 1.0$), but that Leaders tend to converge to low exploration rates quicker than the Followers ($\lambda_L < \lambda_F$). Figure 3*b* shows how different values of $\lambda_L$ impact the observed level of cooperation when $\lambda_F = 0.95$. A nonlinear dependence is obtained, where results show special sensitivity to the overall selection pressure. Notwithstanding, there seems to be an optimal balance in the decay of the exploration rates that allow the population to reach higher levels of cooperation more consistently.

The aforementioned sensitivity to $\beta$ is shown in greater detail in figure 3*c*, where the level of cooperation is depicted as a function of the selection pressure for three values of $\lambda_L$ (with $\lambda_F = 0.95$). When this additional feature is included in the evolutionary dynamics, it becomes possible to attain levels of cooperation of 100%, provided selection pressure is optimally selected.

## 3. Conclusion

The present work investigates in which way both static and time-dependent exploration rates can impact the evolutionary dynamics of cooperation in heterogeneous structured populations. For static and low

exploration rates, we show that the population-wide dynamics no longer resembles the one emerging from the PD game that individuals locally face. Instead, multiple internal roots of the average gradient of selection emerge: two analogues of stable fixed points (probability attractors) separated by one analogue of an unstable fixed point (probability repeller), the latter one inducing a coordination-type dynamics. We also witness the occurrence of a regime shift as we keep increasing the exploration rate, the critical transition taking place at the point where two of the internal roots coalesce and disappear, there remaining a single attractor governing the overall evolutionary dynamics. We find that the critical $\mu$ at which this transition occurs depends on the overall selection pressure.

When the population undergoes a progressive decrease of exploration $\mu$ (from strong to weak), we always observe lower cooperation levels when compared with the situation without any exploration. Similarly to the case of constant exploration rates, we show a non-trivial interplay between exploration rates and selection pressure, and identify an interval of selection pressures that optimizes the resilience of populations against defectors as $\mu$ decreases in time (figure 2).

Such a grim scenario was obtained in situations where all individuals in the population experience the same variations in exploration rates. We find, however, that the overall levels of cooperation may significantly increase provided individuals adopt exploration rates that reflect their role in the underlying social network. At the simplest level, we show that a population with stable Leaders, that is, highly connected individuals that reduce their exploration rate faster than the remainder of their network peers (the Followers), confer to populations a better chance of reaching higher levels of cooperation. For static exploration rates, we show that cooperation can now benefit from exploration (compared with no exploration) if such exploration mostly occurs at the leaves (low connected nodes) of the network. Again, this feature is also sensitive to selection pressure. These results are in accord with those recently proposed by Shirado & Christakis [42], in that the strategic placement of noisy individuals in the right location of social networks can greatly benefit the social goals of a population.

# Methods

## The prisoner's dilemma

Let us assume that individuals may adopt one of two possible behaviours/strategies: to cooperate (C) or to defect (D). Each individual collects a pay-off from each interaction he/she participates in with his/her neighbours. The total pay-off gathered by an individual is computed by

$$\Pi_i = n_i^C(S_i(R - T) + T) + n_i^D(S_i(S - P) + P),\tag{4.1}$$

where $n_i^C$ ($n_i^D$) is the number of cooperators (defectors) in the vicinity of $i$ while $S_i$ is 1 if individual $i$ is a C being 0 when he is a D. The parameters $T$ (temptation), $R$ (reward), $P$ (punishment) and $S$ (sucker's payoff) define the social dilemma faced locally by each individual in a *two-person* and *two-strategy* interaction. It is customary to consider a simplified domain of parameters defined by $R = 1.0$, $P = 0.0$, $-1 \leq S \leq 1.0$ and $0.0 \leq T \leq 2.0$ [29]. Thus, depending on the ordering of the parameters one can define several social dilemmas, of which we shall consider here the most popular: the PD, satisfying the ranking $T > R > P > S$ [31].

## Evolutionary dynamics in well-mixed populations

Let us consider a finite population of size $Z$ and model evolution in discrete time by means of a stochastic birth–death process, in which selection is implemented by the pairwise-comparison rule [13] and strategy exploration is carried out by individuals chosen randomly from the population with a uniform probability $\mu$. The co-evolution of both processes—selection and exploration—can be summarized as follows: at each time step a randomly selected individual, $A$, explores the strategy space by adopting a different strategy (selected at random) with probability $\mu$ or, with probability $1 - \mu$, imitates the strategy of a random neighbour, $B$, with probability

$$p = \frac{1}{1 + e^{-\beta(f_B - f_A)}},\tag{4.2}$$

where $f_i$ denotes the fitness of individual $i$ and $\beta$ the intensity of selection. In well-mixed populations, the dynamics is fully characterized by the gradient of selection (including explorations)

$G_\mu(k) = T_\mu^+(k) - T_\mu^-(k)$, where the $T_\mu^\pm(k)$ are defined as

$$T_\mu^+(k) = (1-\mu)\frac{Z-k}{Z}\frac{k}{Z-1}\frac{1}{1+\exp(-\beta(f_C-f_D))} + \mu\frac{Z-k}{Z} \tag{4.3a}$$

and

$$T_\mu^-(k) = (1-\mu)\frac{k}{Z}\frac{Z-k}{Z-1}\frac{1}{1+\exp(\beta(f_C-f_D))} + \mu\frac{k}{Z} \tag{4.3b}$$

with $T_\mu^+(k)$ referring to the probability of increasing the number of **C**s by one and $T_\mu^-(k)$ to the probability of decreasing the number of **C**s by one, for a given configuration with $k$ **C**s [13]. Using equation (4.3a,b), $G_\mu(k)$ becomes

$$G_\mu(k) = (1-\mu)\frac{Z-k}{Z}\frac{k}{Z-1}\tanh\left(\frac{\beta\Delta f}{2}\right) + \mu\left(1-\frac{2k}{Z}\right) \tag{4.4}$$

where $\Delta f = f_C - f_D$. Whenever this quantity is positive (negative) this means that the number of cooperators is likely to increase (decrease). Inspection of $G_\mu(k)$ reveals that there must exist a critical exploration probability $\mu_C$ above which the second term of equation (4.4) dominates, superseding selection at a population-wide level. Thus, in the extreme scenario of $\mu = 1.0$ the population evolves towards a stationary state in which half of the individuals are cooperators and the remaining half are defectors, resembling a coexistence dynamics characterized by a stable attractor (a finite population analogue of a stable fixed point) located at $x \equiv k/Z = 0.5$. This strong exploration limit is indicated by a blue dashed horizontal line in figure 1. On the other hand, when $\mu = 0.0$, and for PD ($T > R$ and $S < P$) $G_\mu(k)$ is negative for all values of $k$ and defectors are always advantageous, irrespective of their abundance. The inclusion of exploration leads to additional internal roots of $G_\mu(k)$. In the PD case a single stable root ($x_L$) emerges near $x = 0.0$. This root works as an attractor, and moves towards $x = 0.5$ with increasing $\mu$.

## Barabási–Albert algorithm

We generate scale-free networks using an algorithm of growth and preferential attachment proposed by *Barabási and Albert* [30]. In that sense, starting with $m + 1$ fully connected nodes, we iteratively add a new node that connects to $m = 2$ pre-existing nodes proportional to their degree. The algorithm stops when the number of nodes in the network reaches the desired target ($Z$). Networks generated following this procedure will have an average degree of four ($m \times 2$) and a degree distribution that follows a power-law distribution.

## Social network simulations

Each simulation starts with an equal number of Cs and Ds randomly assigned to the nodes of the network. At each time step an individual, $i$, is selected at random from the population. With probability $\mu$, $i$ adopts a different strategy (exploration). With probability $1 - \mu$, $i$ copies the strategy of a random neighbour $j$ with probability given by equation (4.2) (social learning). These steps are repeated for 2.5 million generations (one generation corresponds to $Z$ iterations), or until the population reaches an absorbing state ($k = 0$, or $k = Z$), at which moment the number of cooperators in the population is measured.

*Level of cooperation*, designated by $\eta$, is estimated by averaging the number of cooperators at the end of $10^5$ independent simulations. Each simulation lasts for $5 \times 10^3$ generations and starts from a fraction $x_0$ of Cs that are randomly distributed across the network. Given the coordination nature of the population-wide dynamics in scale-free networks, $\eta$ provides a good approximation for the likelihood that a population reaches a monomorphic state dominated by Cs. Whereas for dynamics dominated by an attractor, it provides a good estimation of its location. Along the manuscript we compute $\eta$ for different scenarios changing exploration levels ($\mu$), selection pressure ($\beta$) and game parameters $T$ and $S$.

*Connectivity classes* are used to split the population between highly connected individuals (Leaders) that have a degree greater than 2/3 of the maximum degree of the population, and lower connected individuals (Followers) that have a degree lower than 2/3 of the maximum degree of the population.

*Variable exploration rates* are implemented by assuming that populations start with the highest possible exploration rates, subsequently decreasing in time geometrically after every predefined time period ($\Delta$):

every $\Delta$ generations (one generation equals $Z$ update events) the exploration rates decay by a constant factor $\lambda$. Thus, the effective exploration rate after $\tau$ time windows is of $\mu_\tau = \mu_0 \lambda^\tau$. We consider two scenarios in the manuscript, when $\lambda$ is the same for all individuals and when it is different depending on the degree/connectivity class to which they belong: $\lambda_L$ for Leaders and $\lambda_F$ for Followers. In the former scenario we consider the case of $\lambda = 0.95$, in the latter scenario we consider scenarios where the Leaders converge to weak exploration levels faster ($\lambda_L < \lambda_F$).

*Average gradient of selection* ($\Gamma_\mu(k, g)$), captures the population-wide dynamics on structured populations and represents the numerical counterpart of $G_\mu(k)$. It can be conveniently estimated by computing the difference between the population averaged probability to increase ($\xi_g^+(k)$) and to decrease ($\xi_g^-(k)$) the number of cooperators by one when the population is in a state with $k$ cooperators (and $Z - k$ defectors). Formally, this quantity is thus defined [11] as

$$\Gamma_\mu(k, g) = \xi_\mu^+(k, g) - \xi_\mu^-(k, g). \tag{4.5}$$

where $\xi_\mu^+(k, g)$ and $\xi_\mu^-(k, g)$, the numerical counterparts of $T_\mu^+(k)$ and $T_\mu^-(k)$, are numerically computed, at the $g$th generation, for exploration probability $\mu$ and for a given configuration with $k$ Cs according to expressions

$$\xi_\mu^-(k, g) = \frac{1}{\Lambda_g(k)} \sum_{\omega=1}^{\Omega} \sum_{t=1}^{t_{max}} \delta(k, k_t) \Theta\left(\frac{t}{Z}, g\right)(1 - \mu)\xi_\omega^-(k, t) + \mu\frac{k}{Z} \tag{4.6a}$$

and

$$\xi_\mu^+(k, g) = \frac{1}{\Lambda_g(k)} \sum_{\omega=1}^{\Omega} \sum_{t=1}^{t_{max}} \delta(k, k_t) \Theta\left(\frac{t}{Z}, g\right)(1 - \mu)\xi_\omega^+(k, t) + \mu\frac{Z - k}{Z}, \tag{4.6b}$$

where $\Lambda_g(k)$ accounts for the total number of times the population was observed in a state with $k$ cooperators at generation $g$ over all $\Omega$ simulations and $\Theta$ $(a, b)$ is a square 'pulse' function that is equal to 1 if $b - 1 \leq a < b$, being 0 otherwise. The first summation (in $\omega$) is over all time-series simulations, whereas the second is over all generations pertaining to each time-series. Finally, $\xi_\omega^\pm(k, t)$ is the transition probability at time $t$ of time-series $\omega$, that is

$$\xi_\omega^\pm(k, t) = \frac{1}{Z} \sum_{i=1}^{Z} \frac{1}{z_i} \sum_{j \in \zeta_i} \frac{1 - \delta(s_j, s_i)}{1 + e^{-\beta(f_j \mp f_i)}}, \tag{4.7}$$

where $s_i$ is 1 (0) if the strategy of individual $i$ is C (D), and the Kronecker $\delta(a, b)$ is equal to 1 if $a = b$ being 0 otherwise. The first summation here is over all individuals in the population, whereas the second spans the entire neighbourhood of each individual. In order to estimate $\Gamma_\mu(k, g)$, we let a population evolve for 150 generations. Each simulation of the evolutionary process starts from an arbitrary random state (i.e. starting from a random number of $k$ cooperators randomly selected from the interval $0 < k < Z$). We repeat this for a total of $\Omega = 2.5 \times 10^7$ times. For each iteration, we estimate the $\xi_\omega^\pm(k, t)$ as described above. In order to characterize the population-wide dynamics we estimate the location of the internal roots of $\Gamma_\mu(k, g)$, which are the states $x^* \approx k^*/Z$ that satisfy $\Gamma_\mu(k^*, g) = 0$. The dynamical role—repeller or attractor—of the internal roots can be assessed by evaluating the sign of $\Gamma_\mu(k, g)$ in its vicinity: for repellers we have that $\Gamma_\mu(k, g) > 0$ for $k < k^*$ and $\Gamma_\mu(k, g) < 0$ for $k > k^*$, and the population will be dynamically driven away from $k^*$ towards $k = 0$ or $k = N$; for attractors we have $\Gamma_\mu(k, g) < 0$ for $k < k^*$ and $\Gamma_\mu(k, g) > 0$ for $k > k^*$, consequently the population will be driven towards $k^*$. Hence, a population is said to be evolving in a coexistence dynamics when its evolutionary outcome is dictated by a single attractor (coexistence point) and characterized by a stable balance of both behaviours in the population. Conversely, in a coordination dynamics the evolutionary outcome is dictated by a single repeller (coordination point), and evolution in the long term leads to the dominance of either behaviour.

Data accessibility. Data used to create the main text Figures and support our conclusions is available for download in the Dryad Digital Repository. A comprehensive description of the standard algorithms implemented to compute the evolutionary dynamics of strategies is provided in the Dryad Digital Repository https://doi.org/10.5061/dryad.m0cfxpp15 [58].
Authors' contributions. F.L.P., J.M.P. and F.C.S. designed and implemented the research, prepared all the figures, wrote and reviewed the manuscript.; F.L.P. conducted the experiment(s).
Competing interests. We declare we have no competing interests.

Funding. This research was supported by Fundação para a Ciência e Tecnologia (FCT) through grant nos. PTDC/MAT/STA/3358/2014, PTDC/MAT-APL/6804/2020, UIDB/04050/2020, UIDB/04152/2020, UIDB/50021/2020 and PTDC/CCI-INF/7366/2020.

Acknowledgements. The authors are thankful to Vítor V. Vasconcelos and Fernando P. Santos for the useful discussions and insights.

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
