## [Peer Review File · Royal Society Open Science]

Review History

RSOS-200910.R0 (Original submission)

Review form: Reviewer 1

Is the manuscript scientifically sound in its present form?

Yes

Are the interpretations and conclusions justified by the results?

Yes

Is the language acceptable?

Yes

Do you have any ethical concerns with this paper?

No

Have you any concerns about statistical analyses in this paper?

No

Recommendation?

Accept with minor revision (please list in comments)

Comments to the Author(s)

This is a neat work in which the authors study the interaction between exploration rates and other stochastic elements of dynamical rules and explore their synergistic impact on cooperation level. As a result, they argue that by allowing leader players to converge faster to low exploration rates can benefit the evolution of cooperative behavior.

This observation is another side of our general understanding about the principal role of heterogeneity when a social dilemma is considered. Notably, the first idea along this avenue was laid by two of the authors many years ago.

I fully agree with the authors that time varying exploration rates are more realistic, but I would like to note that conceptually similar idea has already been raised in PRE 80 (2009) 021901 where time-dependent strategy learning activity was considered.

Let me stress that, however, my note is just optional, I'm happy to support publication of this submission practically in its present form.

Review form: Reviewer 2**Is the manuscript scientifically sound in its present form?**

Yes

Are the interpretations and conclusions justified by the results?

Yes

Is the language acceptable?

Yes

Do you have any ethical concerns with this paper?

No

Have you any concerns about statistical analyses in this paper?

No

Recommendation?

Major revision is needed (please make suggestions in comments)

Comments to the Author(s)

Review for Royal Society Open Science: "Stable leaders pave the way for cooperation under time-dependent exploration rates"

In this paper, the authors analyze what happens to population dynamics when an exploration rate is introduced (additionally to selection via social learning), and what happens if this rate is strong or weak. The authors claim, that in most models studied so far, selection and exploration rates are kept constant, and argue that exploration rates which vary over time are more realistic. It can be expected that individuals who enter a new environment would try out a larger set of strategies to more quickly adapt.

In particular, the authors study structures of populations and argue, that in a heterogeneous social network a prisoners' dilemma type of game might transform to a coordination game which favors cooperation. Indeed, it is well known, that population structure is one of several fundamental mechanisms which promote cooperative behavior. Taking into account mutation (exploration) rates, might make it more difficult for cooperation to sustain, and has already been analyzed in different models. In this paper, the authors, by studying time dependent exploration rates, add another type of feature to the model of evolution of cooperation which might make the situation even more complex.

Comments:

The abstract should be phrased more clearly. From reading the abstract, it is very difficult to understand the point you are trying to make in your paper. Too many different concepts which possibly influence cooperation lead to confusion if you cannot describe your contribution in a better way. You introduce: high/weak exploration rates, high/low selection pressure, high/low levels of cooperation, high/low connectivity in nodes, major differences in collective dynamics. It is hard to follow or understand the point that you are trying to make.

E.g. clarify the sentence: "at intermediate levels of selection...". Rephrase the sentence: "moreover,...".

-On page 2 you write about the differences between theoretical and empirical results for the effects of social networks. Explain in more detail what these differences are.

- I like your idea introduced on page 3 of using different exploration rates for leaders (highly connected individuals) and followers (low connected). But at this point it is not clear if you divide the whole population into these two groups, or if there are more individuals, which do not belong to either of the two extreme groups. Furthermore, you should also write some sentences about the properties of the network that you analyze.

-page 4: you write of an additional internal fixed point. What fixed points do we have so far? What is x ? What is μ ?

Figure 1: From reading the caption, it is impossible for me to understand the graphs. Either you find another way of presenting what you want to show, or you rewrite your captions so that the pictures are understandable. E.g. instead of writing internal roots maybe use the term fixed points? What is x_V , x_L ? Does the white part show that defectors and cooperators coexist, and coordination might be reached as well? What is the co-existence point for well mixed populations? This is all really confusing.

- You write "for $\mu=0.0$ and $\beta=0.01...$ " but I cannot find results for $\mu=0.0$. What do you mean by "a single internal probability repeller"?

- Figure 2 and 3 are easier to understand than figure 1, but still the explanation in the caption could be improved. To my understanding, the difference between figure 2 and 3 is the main point of your paper as it shows the difference between symmetric and asymmetric exploration rates. Maybe you could make that more clear.

- Figure 3: panel A: do followers and leaders just use different exploration rates or is there a time dependence as well? Present panel C such that the differences in time scales between leaders and followers are easier to see.

I did not find an algorithm in the appendix which shows the differentiation between leaders and followers and the set up of the scale free network. Would be interesting to see that.

Minor comments:

Make clear what is meant

- "... these structures..."

- "...connected through an heterogeneous social network... (in a well-mixed sense)..."

- "... such breaks in symmetry..."

- "... in a more positive dynamical scenario"
- "... we probe the impact of variable exploration levels on degree heterogeneous structures populations with a scale-free degree distribution..."

In general the idea of the paper is interesting, and your analysis and model is sound and well done. However, major parts of the paper are difficult to read and understand, and your figures could be improved. In that way, it is very difficult to assess what the contribution of your paper actually is, and what the introduction of time dependent exploration rates and the distinction of leaders and followers in terms of their exploration rates really means. What happens if exploration rates increase over time? What happens if other assumptions that you make are relaxed? If you can present the most important results of your paper in a better way, this could make a good contribution to the field. However, in the present form I suggest that you revise your manuscript.

Decision letter (RSOS-200910.R0)

Dear Dr Pinheiro,

The editors assigned to your paper ("Stable leaders pave the way for cooperation under time-dependent exploration rates") have now received comments from reviewers. We would like you to revise your paper in accordance with the referee and Associate Editor suggestions which can be found below (not including confidential reports to the Editor). Please note this decision does not guarantee eventual acceptance.

Please submit a copy of your revised paper before 29-Aug-2020. Please note that the revision deadline will expire at 00.00am on this date. If we do not hear from you within this time then it will be assumed that the paper has been withdrawn. In exceptional circumstances, extensions may be possible if agreed with the Editorial Office in advance. We do not allow multiple rounds of revision so we urge you to make every effort to fully address all of the comments at this stage. If deemed necessary by the Editors, your manuscript will be sent back to one or more of the original reviewers for assessment. If the original reviewers are not available, we may invite new reviewers.

- Data accessibility

<http://datadryad.org/submit?journalID=RSOS&manu=RSOS-200910>

- Competing interests

- Authors' contributions

- Acknowledgements

- Funding statement

on behalf of Dr Mirco Musolesi (Associate Editor) and Marta Kwiatkowska (Subject Editor)
openscience@royalsociety.org

Associate Editor's comments (Dr Mirco Musolesi):

Associate Editor: 1

Comments to the Author:

The reviewers are quite positive about this paper, but major changes are needed before accepting it for publication.

We would like to invite the authors to address the concerns of the reviewers carefully. One of the reviewers also identified various presentation issues, which should be addressed by the authors in their revision.

Comments to Author:

Reviewers' Comments to Author:

Reviewer: 1

Comments to the Author(s)

This is a neat work in which the authors study the interaction between exploration rates and other stochastic elements of dynamical rules and explore their synergistic impact on cooperation level. As a result, they argue that by allowing leader players to converge faster to low exploration rates can benefit the evolution of cooperative behavior.

This observation is another side of our general understanding about the principal role of heterogeneity when a social dilemma is considered. Notably, the first idea along this avenue was laid by two of the authors many years ago.

I fully agree with the authors that time varying exploration rates are more realistic, but I would like to note that conceptually similar idea has already been raised in PRE 80 (2009) 021901 where time-dependent strategy learning activity was considered.

Let me stress that, however, my note is just optional, I'm happy to support publication of this submission practically in its present form.

Reviewer: 2

Comments to the Author(s)

Review for Royal Society Open Science: "Stable leaders pave the way for cooperation under time-dependent exploration rates"

In this paper, the authors analyze what happens to population dynamics when an exploration rate is introduced (additionally to selection via social learning), and what happens if this rate is strong or weak. The authors claim, that in most models studied so far, selection and exploration rates are kept constant, and argue that exploration rates which vary over time are more realistic. It can be expected that individuals who enter a new environment would try out a larger set of strategies to more quickly adapt.

In particular, the authors study structures of populations and argue, that in a heterogeneous social network a prisoners' dilemma type of game might transform to a coordination game which

favors cooperation. Indeed, it is well known, that population structure is one of several fundamental mechanisms which promote cooperative behavior. Taking into account mutation (exploration) rates, might make it more difficult for cooperation to sustain, and has already been analyzed in different models. In this paper, the authors, by studying time dependent exploration rates, add another type of feature to the model of evolution of cooperation which might make the situation even more complex.

Comments:

The abstract should be phrased more clearly. From reading the abstract, it is very difficult to understand the point you are trying to make in your paper. Too many different concepts which possibly influence cooperation lead to confusion if you cannot describe your contribution in a better way. You introduce: high/weak exploration rates, high/low selection pressure, high/low levels of cooperation, high/low connectivity in nodes, major differences in collective dynamics. It is hard to follow or understand the point that you are trying to make.

E.g. clarify the sentence: "at intermediate levels of selection...". Rephrase the sentence: "moreover,...".

-On page 2 you write about the differences between theoretical and empirical results for the effects of social networks. Explain in more detail what this differences are.

- I like your idea introduced on page 3 of using different exploration rates for leaders (highly connected individuals) and followers (low connected). But at this point it is not clear if you divide the whole population into these two groups, or if there are more individuals, which do not belong to either of the two extreme groups. Furthermore, you should also write some sentences about the properties of the network that you analyze.

-page 4: you write of an additional internal fixed point. What fixed points do we have so far? What is x ? What is μ ?

Figure 1: From reading the caption, it is impossible for me to understand the graphs. Either you find another way of presenting what you want to show, or you rewrite your caption s that the pictures are understandable. E.g. instead of writing internal roots maybe use the term fixed points? What is x_V , x_L ? Does the white part show that defectors and cooperators coexist, and coordination might be reached as well? What is the co-existence point for well mixed populations? This is all really confusing.

- You write "for $\mu=0.0$ and $\beta=0.01$..." but I cannot find results for $\mu=0.0$. What do you mean by "a single internal probability repeller"?

- Figure 2 and 3 are easier to understand than figure 1, but still the explanation in the caption could be improved. To my understanding, the difference between figure 2 and 3 is the main point of your paper as it shows the difference between symmetric and asymmetric exploration rates. Maybe you could make that more clear.

- Figure3: panel A: do followers and leaders just use different exploration rates or is there a time dependence as well? Present panel C such that the differences in time scales between leaders and followers are easier to see.

I did not find an algorithm in the appendix which shows the differentiation between leaders and followers and the set up of the scale free network. Would be interesting to see that.

Minor comments:

Make clear what is meant

- "... these structures..."

- "...connected through an heterogeneous social network... (in a well-mixed sense)..."

- "... such breaks in symmetry..."

- "... in a more positive dynamical scenario"

- "... we probe the impact of variable exploration levels on degree heterogeneous structures populations with a scale-free degree distribution..."

In general the idea of the paper is interesting, and your analysis and model is sound and well done. However, major parts of the paper are difficult to read and understand, and your figures could be improved. In that way, it is very difficult to assess what the contribution of your paper actually is, and what the introduction of time dependent exploration rates and the distinction of leaders and followers in terms of their exploration rates really means. What happens if exploration rates increase over time? What happens if other assumptions that you make are relaxed? If you can present the most important results of your paper in a better way, this could make a good contribution to the field. However, in the present form I suggest that you revise your manuscript.

Author's Response to Decision Letter for (RSOS-200910.R0)

See Appendix A.

RSOS-200910.R1 (Revision)

Review form: Reviewer 1

Is the manuscript scientifically sound in its present form?

Yes

Are the interpretations and conclusions justified by the results?

Yes

Is the language acceptable?

Yes

Do you have any ethical concerns with this paper?

No

Have you any concerns about statistical analyses in this paper?

No

Recommendation?

Accept as is

Comments to the Author(s)

I think the authors made a good job, the revised version is even more enjoyable than the original one. I'm happy to recommend publication!

Review form: Reviewer 2

Is the manuscript scientifically sound in its present form?

Yes

Are the interpretations and conclusions justified by the results?

Yes

Is the language acceptable?

Yes

Do you have any ethical concerns with this paper?

No

Have you any concerns about statistical analyses in this paper?

No

Recommendation?

Accept with minor revision (please list in comments)

Comments to the Author(s)

Review of revised manuscript for Royal Society Open Science: "Stable leaders pave the way for cooperation under time-dependent exploration rates"

I acknowledge that the authors put effort in making their points more clear. However, some of my questions/concerns are still not answered.

Comments:

Confusing: "...the positive impact on cooperation of structured population.", as you mean "... the positive impact of structured population on cooperation".

Unclear: "depending on their degree" (the notion degree is not clear here)

It is perfectly ok to write internal roots, but on page 10 you fail to provide a definition of x_L , x and x_U . I told you that in my last review, now you moved it to the main text, but it is still not clear. I understand that the dynamics is just about the frequency of cooperators, x , and that there is a fixed point with low frequency and one with high. But that's not clear from you text, you should explain and define that. What do you mean by a co-existence game? If $x=0.5$, so 50% are cooperators, you call that a coexistence game? Did you want to say coordination game? Unclear. Again, as already stated in my last review, what do you mean by "a single internal probability repeller at x^* " you should use a different formulation for what you want to say

The same with "internal probability attractor". It would help if you would at least cancel the word "probability".

You write that for some critical μ values two internal roots coalesce. Do they disappear completely if μ is further increased? That's not clear from the figure.

What is the relative level of cooperation that you depict in figure 3, compared to level of cooperation in figure 2?

Decision letter (RSOS-200910.R1)

Dear Dr Pinheiro

On behalf of the Editors, we are pleased to inform you that your Manuscript RSOS-200910.R1 "Stable leaders pave the way for cooperation under time-dependent exploration rates" has been accepted for publication in Royal Society Open Science subject to minor revision in accordance with the referees' reports. Please find the referees' comments along with any feedback from the Editors below my signature.

Please submit your revised manuscript and required files (see below) no later than 7 days from today's (ie 30-Nov-2020) date. Note: the ScholarOne system will 'lock' if submission of the revision is attempted 7 or more days after the deadline. If you do not think you will be able to meet this deadline please contact the editorial office immediately.

on behalf of Dr Mirco Musolesi (Associate Editor) and Marta Kwiatkowska (Subject Editor)
openscience@royalsociety.org

Associate Editor Comments to Author (Dr Mirco Musolesi):

Associate Editor: 1

Comments to the Author:

The authors addressed all the major concerns of the reviewers. The authors should address the remaining comments of one of the reviewers before the final acceptance of the manuscript.

Reviewer comments to Author:

Reviewer: 1

Comments to the Author(s)

I think the authors made a good job, the revised version is even more enjoyable than the original one. I'm happy to recommend publication!

Reviewer: 2

Comments to the Author(s)

Review of revised manuscript for Royal Society Open Science: "Stable leaders pave the way for cooperation under time-dependent exploration rates"

I acknowledge that the authors put effort in making their points more clear. However, some of my questions/concerns are still not answered.

Comments:

Confusing: “..the positive impact on cooperation of structured population.”, as you mean “.. the positive impact of structured population on cooperation”.

Unclear: “ depending on their degree” (the notion degree is not clear here)

It is perfectly ok to write internal roots, but on page 10 you fail to provide a definition of x_L , x and x_U . I told you that in my last review, now you moved it to the main text, but it is still not clear. I understand that the dynamics is just about the frequency of cooperators, x , and that there is a fixed point with low frequency and one with high. But that's not clear from you text, you should explain and define that. What do you mean by a co-existence game? If $x=0.5$, so 50% are cooperators, you call that a coexistence game? Did you want to say coordination game? Unclear. Again, as already stated in my last review, what do you mean by “ a single internal probability repeller at x^* ” you should use a different formulation for what you want to say

The same with “internal probability attractor”. It would help if you would at least cancel the word “probability”.

You write that for some critical μ values two internal roots coalesce. Do they disappear completely if μ is further increased? That's not clear from the figure.

What is the relative level of cooperation that you depict in figure 3, compared to level of cooperation in figure 2?

===PREPARING YOUR MANUSCRIPT===

===PREPARING YOUR REVISION IN SCHOLARONE===

Author's Response to Decision Letter for (RSOS-200910.R1)

See Appendix B.

Decision letter (RSOS-200910.R2)

Dear Dr Pinheiro,

It is a pleasure to accept your manuscript entitled "Stable leaders pave the way for cooperation under time-dependent exploration rates" in its current form for publication in Royal Society Open Science.

on behalf of Dr Mirco Musolesi (Associate Editor) and Marta Kwiatkowska (Subject Editor)

Appendix A

Lisboa, August 28th, 2020

Dear Prof. Mirco Musolesi,

Thank you for the opportunity to revise and resubmit our manuscript. We are very happy with the overall positive assessment of our manuscript by both Referees. We are also grateful for all the comments and suggestions raised, which helped us to improve the overall quality of our contribution.

Enclosed, please find a point-by-point answer to all comments and concerns raised by the Reviewers. We have carefully revised the manuscript accordingly and hope this new version meets the stringent quality requirements of Royal Society Open Science. Particularly, we hope to have satisfactorily addressed all the concerns raised by Reviewer #2, and to properly contextualize the reference suggested by Reviewer #1.

For your convenience, we attach a PDF version of our contribution highlighting all the changes made to our previous submission.

With best regards,

Flávio L. Pinheiro, Jorge M. Pacheco, and Francisco C. Santos

Reviewer #1

This is a neat work in which the authors study the interaction between exploration rates and other stochastic elements of dynamical rules and explore their synergistic impact on cooperation level. As a result, they argue that by allowing leader players to converge faster to low exploration rates can benefit the evolution of cooperative behavior.

A: We would like to thank Reviewer #1 for his/her time reviewing our manuscript and kind words of appreciation. We are very happy with your assessment.

This observation is another side of our general understanding about the principal role of heterogeneity when a social dilemma is considered. Notably, the first idea along this avenue was laid by two of the authors many years ago.

I fully agree with the authors that time varying exploration rates are more realistic, but I would like to note that conceptually similar idea has already been raised in PRE 80 (2009) 021901 where time-dependent strategy learning activity was considered.

Let me stress that, however, my note is just optional, I'm happy to support publication of this submission practically in its present form.

A: We thank the Referee for calling our attention to this Reference, that we now cite, along with a short discussion of the literature.

Reviewer #2

In this paper, the authors analyze what happens to population dynamics when an exploration rate is introduced (additionally to selection via social learning), and what happens if this rate is strong or weak. The authors claim, that in most models studied so far, selection and exploration rates are kept constant, and argue that exploration rates which vary over time are more realistic. It can be expected that individuals who enter a new environment would try out a larger set of strategies to more quickly adapt.

In particular, the authors study structures of populations and argue, that in a heterogeneous social network a prisoners' dilemma type of game might transform to a coordination game which favors cooperation. Indeed, it is well known, that population structure is one of several fundamental mechanisms which promote cooperative behavior. Taking into account mutation (exploration) rates, might make it more difficult for cooperation to sustain, and has already been analyzed in different models. In this paper, the authors, by studying time dependent exploration rates, add another type of feature to the model of evolution of cooperation which might make the situation even more complex.

A: We would like to thank Reviewer #2 for his/her time reviewing our manuscript. We acknowledge the concerns of the reviewer with respect to the readability of the manuscript. In that sense, we have prepared a revised version that addresses his/her main concerns. Please find below a point-by-point answer to all comments and questions.

Comments:

The abstract should be phrased more clearly. From reading the abstract, it is very difficult to understand the point you are trying to make in your paper. Too many different concepts which possibly influence cooperation lead to confusion if you cannot describe your contribution in a better way.

You introduce: high/weak exploration rates, high/low selection pressure, high/low levels of cooperation, high/low connectivity in nodes, major differences in collective dynamics. It is hard to follow or understand the point that you are trying to make. E.g. clarify the sentence: "at intermediate levels of selection... ". Rephrase the sentence: "moreover,...".

A: Thank you. We have rewritten the abstract in order to address the concerns of the reviewer. The new version is shorter and goes directly to the main topic, avoiding possible sources of confusion. Some of the terms originally included in the abstract are now deferred to the main text. We hope the reviewer agrees with us that the new version is clearer, cleaner, and summarizes the main aspects of our contribution.

-On page 2 you write about the differences between theoretical and empirical results for the effects of social networks. Explain in more detail what this differences are.

A: We have rewritten the paragraph and we hope the reviewer agrees with us that it is now clearer.

- I like your idea introduced on page 3 of using different exploration rates for leaders (highly connected individuals) and followers (low connected). But at this point it is not clear if you divide the whole population into these two groups, or if there are more individuals, which do not belong to either of the two extreme groups. Furthermore, you should also write some sentences about the properties of the network that you analyze.

A: Thank you. We have added a paragraph in the Methods section that describes the algorithm of growth and preferential attachment (also commonly known as Barabási-Albert algorithm) used to generate the networks. Moreover, we added a reference to the Methods section in page 3 where the algorithm is detailed.

-page 4: you write of an additional internal fixed point. What fixed points do we have so far? What is x ? What is μ ?

A: We have clarified this information both in the Caption of Figure 1 and we in the main text.

Figure 1: From reading the caption, it is impossible for me to understand the graphs. Either you find another way of presenting what you want to show, or you rewrite your caption so that the pictures are understandable. E.g. instead of writing internal roots maybe use the term fixed points? What is x_V , x_L ? Does the white part show that defectors and cooperators coexist, and coordination might be reached as well? What is the co-existence point for well mixed populations? This is all really confusing.

A: Thank you for pointing this out. Mathematical rigor dictates that fixed points are realized in deterministic dynamics taking place in infinite populations. In finite populations undergoing stochastic dynamics, the designation "fixed-points" is abusive, despite its frequent usage. Here we opted to stick to a designation that can be easily understood if we use as a reference scenario that of infinite well-mixed populations discussed in the Methods section. Not only did we rewrite the Figure caption taking your comments into consideration, that is, making the interpretation of the "fixed-points" easier to understand, but we also added a reference to the Methods in Figure 1.

- You write "for $\mu=0.0$ and $\beta=0.01$..." but I cannot find results for $\mu=0.0$. What do you mean by "a single internal probability repeller"?

A: Thank you. Results for " $\mu = 0.0$ " are shown in the left border panels A, B and C of Figure 1. They were overlaid on top of this border since we are using a logarithm scale in the X axis. We have clarified this in caption to Figure 1. (see also previous answer).

- Figure 2 and 3 are easier to understand than figure 1, but still the explanation in the caption could be improved. To my understanding, the difference between figure 2 and 3 is the main point of your paper as it shows the difference between symmetric and asymmetric exploration rates. Maybe you could make that more clear.

A: Thank you for pointing this out. We added a sentence in Figure 3 that clarifies that exploration rates in Panel A and B are constant and set at the start of the simulations, while for Panel C they are time-dependent.

- Figure3: panel A: do followers and leaders just use different exploration rates or is there a time dependence as well? Present panel C such that the differences in time scales between leaders and followers are easier to see.

A: We believe that the changes made do overcome your objections. Thank you for raising our attention to this potential misunderstanding regarding Fig. 3.

I did not find an algorithm in the appendix which shows the differentiation between leaders and followers and the set up of the scale free network. Would be interesting to see that.

A: Given the previous answers and the changes introduced in the main text, we believe it is clear that no changes in the algorithm are necessary - only the time scales associated with leaders and followers. In what concerns the algorithm to generate scale-free networks, it is the well-known growth and preferential attachment algorithm now discussed in detail in the Methods section, and by the original authors in References 31 and 59 of revised version

Minor comments:

Make clear what is meant

- "... these structures..."
- "...connected through an heterogeneous social network... (in a well-mixed sense)..."
- "... such breaks in symmetry..."
- "... in a more positive dynamical scenario"
- "... we probe the impact of variable exploration levels on degree heterogeneous structures populations with a scale-free degree distribution..."

As: Thank you. We have reworded these sentences to make them clear.

In general the idea of the paper is interesting, and your analysis and model is sound and well done. However, major parts of the paper are difficult to read and understand, and your figures could be improved. In that way, it is very difficult to assess what the contribution of your paper actually is, and what the introduction of time dependent exploration rates and the distinction of leaders and followers in terms of their exploration rates really means. What happens if exploration rates increase over time? What happens if other assumptions that you make are relaxed? If you can present the most important results of your paper in a better way,

this could make a good contribution to the field. However, in the present form I suggest that you revise your manuscript.

A: We hope the reviewer finds our modifications satisfactory, addressing all of his/her concerns regarding the readability of our manuscript.

Once again, we would like to thank the reviewer for all comments and suggestions. It definitely helped us to improve our manuscript. Thank you very much.

Appendix B

Lisbon, January 8, 2021

Dear Prof. Mirco Musolesi and Marta Kwiatkowska,

Thank you for your time handling our manuscript review process. We are very pleased with the acceptance of our manuscript.

Enclosed please find a point-by-point answer to all comments and concerns raised by the Reviewers. We have carefully revised the manuscript taking them into full consideration. Thus, we hope this new version meets the stringent quality requirements of Royal Society Open Science. More specifically, the revised manuscript addresses the concerns raised by Reviewer #2 and adds the reference suggested by Reviewer #1 along with a discussion of the literature. Enclosed we also provide a pdf that highlights/tracks all the changes made to our previous submission. Thank you again for all your help and support.

All our best wishes for 2021!

With best regards,
Flávio L. Pinheiro, Jorge M. Pacheco, and Francisco C. Santos

Editor #1

The authors addressed all the major concerns of the reviewers. The authors should address the remaining comments of one of the reviewers before the final acceptance of the manuscript.

A: We would like to thank the Editors for taking their time handling our submission. We are very pleased with the acceptance of our manuscript and thankful for the opportunity to further revise it and, this way, address all reviewers' comments which, once again, helped us to improve our contribution.

Reviewer #1

I think the authors made a good job, the revised version is even more enjoyable than the original one. I'm happy to recommend publication!

A: We would like to thank Reviewer #1 for his/her time reviewing our manuscript and his/her positive assessment of our contribution. Thank you!

Reviewer #2

I acknowledge that the authors put effort in making their points more clear. However, some of my questions/concerns are still not answered.

A: We would like to thank Reviewer #2 for his/her time reviewing our manuscript and his/her positive assessment of the changes we introduced in the last resubmission of our manuscript. Thank you so much for your time. Below please find point-by-point answers to all your comments and concerns.

Comments:

Confusing: "...the positive impact on cooperation of structured population.", as you mean "... the positive impact of structured population on cooperation".

A: Thank you. We rephrased the sentence to include "the positive impact of population structure in the evolution of cooperation".

Unclear: " depending on their degree" (the notion degree is not clear here)

A: Thank you for raising our attention to this point. We fully agree. We replaced degree by "number of contacts" of an individual, clarifying the intended meaning.

It is perfectly ok to write internal roots, but on page 10 you fail to provide a definition of x_L , x and x_U . I told you that in my last review, now you moved it to the main text, but it is still not clear. I understand that the dynamics is just about the frequency of cooperators, x , and that there is a fixed point with low frequency and one with high. But that's not clear from you text, you should explain and define that.

What do you mean by a co-existence game? If $x=0.5$, so 50% are cooperators, you call that a coexistence game? Did you want to say coordination game? Unclear.

A: We hope to have finally clarified the definitions of co-existence and coordination games in the methods, and how they are linked to the different nature of the internal roots (repeller and attractor) that characterize the stochastic gradient of selection.

Again, as already stated in my last review, what do you mean by " a single internal probability repeller at x^* " you should use a different formulation for what you want to say
The same with "internal probability attractor". It would help if you would at least cancel the word "probability".

A: Thank you. We have eliminated, whenever possible, the word probability in "internal probability repeller" and "internal probability attractor".

You write that for some critical μ values two internal roots coalesce. Do they disappear completely if μ is further increased? That's not clear from the figure.

A: We clarified this in text. The coalescence always happens for increasing μ as long as the gradient of selection for $\mu \sim 0$ is characterized by a coordination point. The value of μ at which the internal roots coalesce is however dependent on the remaining parameters. We have also clarified this in the main text.

What is the relative level of cooperation that you depict in figure 3, compared to level of cooperation in figure 2?

A: In both figures the level of cooperation was estimated in a similar manner; the main difference has to do with the conditions that were tested. In Figure 2 we focus in the role of Selection Pressure and Decay Time Scales, while in Figure 3 we explore the role of asymmetric exploration rates between Leaders and Followers. We explicitly mention this in the newly revised version.

Once again, we want to thank Reviewer #2 for his/her feedback, which allowed us to improve the clarity of our manuscript rendering it more accessible to a broader audience.